# ROS-PyBullet Interface: A Framework for Reliable Contact Simulation and Human-Robot Interaction

**Christopher E. Mower**[*,1,3], **Theodoros Stouraitis**[*,2,3], **João Moura**[3,5], **Christian Rauch**[3],
**Lei Yan**[4], **Nazanin Zamani Behabadi**, **Michael Gienger**[2], **Tom Vercauteren**[1],
**Christos Bergeles**[1], **and Sethu Vijayakumar**[3,5]

[1]King's College London, London, UK. [2]Honda Research Institute, Offenbach, Germany.
[3]The University of Edinburgh, Edinburgh, UK. [4]Harbin Institute of Technology,
Shenzhen, China. [5]The Alan Turing Institute, London, UK.
`christopher.mower@kcl.ac.uk, theostou@honda-ri.de`

**Abstract:** Reliable contact simulation plays a key role in the development of (semi-)autonomous robots, especially when dealing with contact-rich manipulation scenarios, an active robotics research topic. Besides simulation, components such as sensing, perception, data collection, robot hardware control, human interfaces, etc. are all key enablers towards applying machine learning algorithms or model-based approaches in real world systems. However, there is a lack of software connecting reliable contact simulation with the larger robotics ecosystem (i.e. ROS, Orocos), for a more seamless application of novel approaches, found in the literature, to existing robotic hardware. In this paper, we present the ROS-PyBullet Interface, a framework that provides a bridge between the reliable contact/impact simulator PyBullet and the Robot Operating System (ROS). Furthermore, we provide additional utilities for facilitating Human-Robot Interaction (HRI) in the simulated environment. We also present several use-cases that highlight the capabilities and usefulness of our framework. Our code base is open source and can be found at github.com/ros-pybullet/ros_pybullet_interface.

## 1 Introduction

Dealing with contacts is a key requirement for robots to become effective in our daily lives and valuable assets in industry [1]. Examples of contact-rich tasks include pick and place [2], locomotion [3, 4, 5], wiping [6], pushing [7, 8], dyadic co-manipulation [9], and robot surgery [10, 11, 12, 13]. Developing approaches for (semi-)autonomous robots involving contact is thwart with practical issues (e.g. slippage, high impulsive forces, model mismatch, etc.) that may cause failure and damage to the robot or yield safety concerns for humans in close proximity.

Development of machine learning approaches require a facility to collect large datasets. However, collection on scale is non-trivial. Learning from demonstration [14] is a popular technique for endowing robots with new skills where a human provides examples. According to one paradigm, *kinaesthetic teaching* (e.g. [15]), the human interacts directly with the robot. However, this method requires a physical system which is cumbersome and has potential safety concerns. A second approach utilizes *teleoperation* (e.g. [16]), which has the benefit that the human operator can either interact with a simulated robot or at a distance with the physical system (ensuring safety). Two key considerations for our framework are that the virtual world can include a human model via telepresence and that the human operator can experience the virtual forces generated by the simulator, through interface to haptic devices. Furthermore, to ease issues when deploying methods on robot hardware, we provide software features to easily map targets to the robot control commands.

In robotics, the system's complexity, need for several sub-processes, and multi-machine operations motivate a modular system design and message parsing functionality [17, 18, 19]. There are many packages for the Robot Operating System (ROS) [18] integrating useful functionalities for research and commercial systems, including useful data structures, control interfaces, inverse kinematics (IK) and motion planning, perception tools, etc. [20, 21, 22, 23, 24]. Additionally, various visual-

---

[*]The majority of this work was done while at The University of Edinburgh.

6th Conference on Robot Learning (CoRL 2022), Auckland, New Zealand.

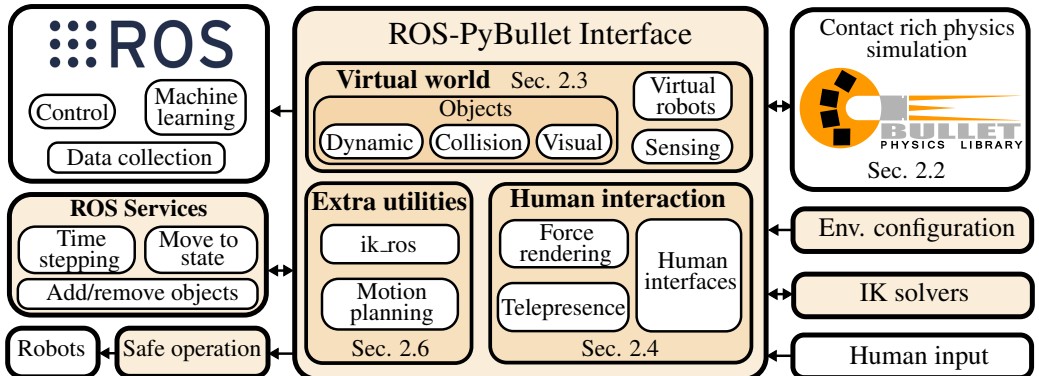

Figure 1: Outline of our framework for bridging a reliable contact simulator within the ROS ecosystem, and HRI interfaces. Colored boxes indicate the specific parts of our proposed framework.

izers [25, 26, 27] and physics simulators [28, 29, 30, 31, 32] allow for the development and testing of algorithms, prior to execution on robot hardware. However, popular and reliable libraries for contact-rich scenarios, such as PyBullet [31], Drake [30], and MuJoCo [29], lack integration with the ROS framework.

## 1.1 Contributions

In this paper, our contributions are

- A framework (Figure 1) that enables research in contact-rich manipulation scenarios allowing for seamless collection of contact-rich data and human demonstrations within a simulated environment.
- Our system enables transference of robot behaviors (learned or otherwise) from the PyBullet simulation environment (a reliable contact/impact simulator) to hardware using ROS.
- Implementation of several HRI interfaces, e.g. keyboard, mouse, joystick, 6D-mouse, haptic devices, Xsens suit that enable easy interaction with virtual environments for HRI and haptic setups.
- Several use-cases to demonstrate the capabilities and usefulness of the framework, including a variety of robots (e.g. Kawada Nextage humanoid, KUKA LWR robot arm, Kinova arm, and dual-arm KUKA IIWA), and full documentation (see supplementary material).

Our framework exploits modularity and extensibilty by implementing several ROS nodes and object types using class hierarchy design patterns. Furthermore, we include additional features such as: utilities for Model Predictive Control (MPC) design and development, motion planning, safe robot operation (for ensuring safety limits are satisfied before commanding the real robot), and inverse kinematics. Several recent works use the proposed framework [6, 33, 8, 34, 35].

## 2 Proposed Framework

To enable easy prototyping, implementation, and integration with hardware we implement several tools/features in the framework. This section describes these and our design decisions.

## 2.1 Framework features

Figure 1 shows an overview for our framework highlighting it as a central interface between the ROS ecosystem and PyBullet, human interaction, IK solvers, and real robots. The main features of the framework are listed as follows.

1. **Online, full-physics simulation using a reliable contact simulator.** The framework relies on PyBullet to enable well established contact simulation for rigid/deformable bodies.
2. **Integration with the ROS ecosystem.** Robot simulation and visualization of real robots/objects (utilizing sensing) are integrated via ROS. Furthermore, this enables (i) ROS packages to be integrated with PyBullet and (ii) a straightforward way to port developed algorithms to real systems.
3. **Several interfaces enabling HRI with virtual worlds and telepresence.** We provide facilities for human's to provide examples in a simulated environment via several popular interfaces (including haptic devices).

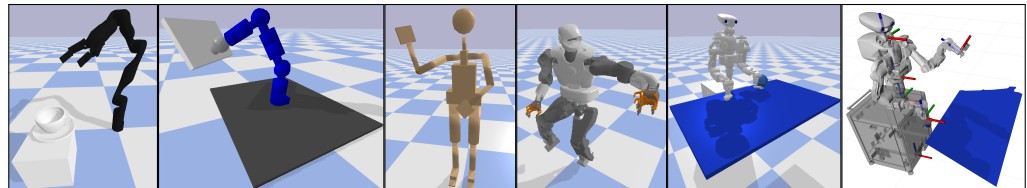

Figure 2: Examples of the ROS-PyBullet Interface: (left-right): a Kinova arm reaching for a cup, a Kuka LWR arm simulating a wiping task, a human model waving, a Talos humanoid robot reaching for a target while maintaining balance, the Nextage robot simulating a smart factory scenario, and simulated RGB-D data visualized in RViz.

4. **Sensor simulation.** Robot joint force-torque sensors and RGB-D cameras (i.e. point clouds) are provided for sensing-based control and can be seamlessly interchanged with real world sensor streams via ROS.

5. **Modular and extensible design.** Our framework adopts a modular (i.e. several ROS nodes) and highly extensible design paradigm (i.e. class hierarchy) using the Python programming language. This makes it easy to quickly develop new features for the framework.

6. **Data collection with standard ROS tools.** Since the framework provides an interface to ROS, we can leverage common tools for data collection such as ROS bags [36] and data processing to common formats in machine learning applications, i.e. `rosbag_pandas` [37].

7. **Integration with robot and sensing hardware.** Tools are provided to easily remap the virtual system to physical hardware and integrate real sensing apparatus in the PyBullet simulation (e.g. vicon).

## 2.2 Full-Physics Contact-rich Simulation

Several simulators exist for full-physics simulation e.g. Gazebo [27], ODE [28], Nvidia Isaac [38], DART [39], PyBullet [31], Drake [30], and MuJoCo [29]. We chose to use PyBullet [31] since it is free and open source, a well-known library with an active community, easy to install, well documented, and is in Python. A more detailed comparison is given in Section 4. To ensure our framework is extensible, we develop several classes that interface with PyBullet and establish communication links with ROS utilizing publishers, subscribers, timers, and services.

## 2.3 Robots and virtual worlds building

Our framework provides several tools to build virtual worlds. The main ROS-PyBullet Interface node must specify a main YAML configuration file containing a list of robots/objects to load into PyBullet, parameters, RGB-D sensor configuration, and visualizer options - robots/objects can also be added/removed through ROS services. Various examples of robots/tasks are shown in Figure 2. Several object types (Section 2.3.1) were developed with different interaction properties and various communication channels with ROS. The specification for each object is defined in separate YAML files. See the documentation provided in the supplementary material for a full list of parameters for the configuration files, ROS topics published/subscribed, and ROS services provided.

### 2.3.1 PyBullet Objects

**Robot** Incorporating robots in our framework is simple. In a configuration file the user will specify the URDF file name, and several parameters (e.g. base position, inital joint configuration, etc). The robot base frame with respect to the world frame (named `rpbi/world`) is set by a transform broadcast using the ROS TF library [20]. The interface optionally publishes the robot joint/link states to ROS. Mobile and floating-base robots are also supported by our framework given an initial state (i.e. pose and linear/angular velocity). ROS services expose PyBullet's IK features, allow the user to move the the robot to a given joint/end-effector state[2], and return robot information (e.g. joint/link names, number of degrees of freedom, etc).

**Visual robot** A visualization of a robot, that does not interact with the PyBullet environment, can be instantiated by setting the `is_visual_robot` parameter to `True` in the robot configuration file. The main utility of this feature is to allow the user to map a real robot state to the PyBullet environment enabling them to easily compare the real robot with a simulated robot representing the target configuration. Only the robot information and IK services are available for visual robots.

---

[2] When an end-effector state is given, the corresponding joint state is found using PyBullet's IK features.

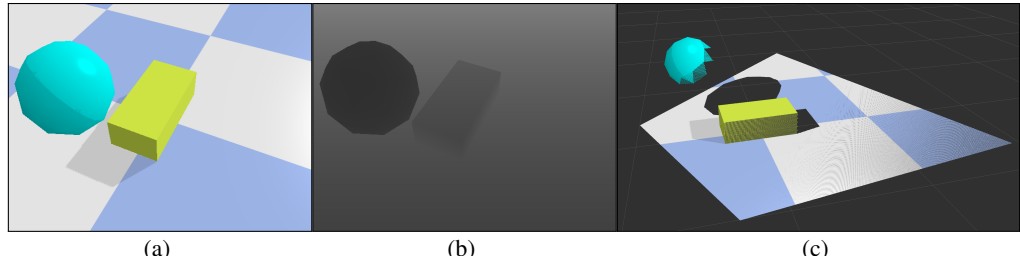

|  (a)  |  (b)  |  (c)  |

Figure 3: RGB-D observation of a scene with two objects: (a) colour image, (b) depth image, (c) projected colour point cloud in RViz.

**Visual object** A common requirement for simulators is to visualize objects. In PyBullet, these objects do not affect other bodies in the scene, nor react to those. Visual objects are listed in the main configuration file under the parameter `visual_objects`. A visual object was used to visualize the real pushing box (tracked with Vicon) in [8].

**Collision object** Modeling static objects, that cause momentum changes for other bodies upon collision, such as floors, ceilings, and walls are often necessary. These objects are listed under the parameter `collision_objects` in the main configuration. During the development of the experiments presented in [6], a collision object was used to represent a surface.

**Dynamic object** Objects whose pose evolution is completely determined by the simulator's physics engine are a key requirement for development of control algorithms. Dynamic objects, listed as `dynamic_objects` in the main configuration file, can be used inside PyBullet to simulate an object. A simulated pushing box was used to develop the controller in [8].

**Soft object** All previously described object types are rigid bodies. We also provide an interface to PyBullet deformable objects. These are similar to dynamic objects in that their evolution is defined by PyBullet and can be specified using the `soft_objects` parameter.

**Load from URDF** Finally, robots and objects can also be loaded directly into the PyBullet environment using the `urdfs` parameter. The evolution of these objects are defined by PyBullet. However, since the usage of these objects is ambiguous their communication with ROS is limited.

### 2.3.2 Sensor simulation

Many control, and planning algorithms rely on sensory feedback. Integrating multiple sensory inputs, such as tactile and vision, is underdeveloped in robotic manipulation [40]. To enable future research in realistic simulated environments, we provide an interface to several sensing modalities.

**F/T sensor** Through the configuration file for the robot it is possible to instantiate a simulated force-torque sensor attached to any joint on the robot. These virtual sensors publish joint reaction forces, read from PyBullet, as ROS wrench-stamped messages at a user-defined sampling frequency.

**RGB-D camera** Color and depth perception is a key sensing capability for object state estimation in contact-rich manipulation tasks. An RGB-D camera can be instantiated and attached to a frame through the main configuration file.

The color and depth images (`image`) are published together with the intrinsic camera parameters (`camera_info`). The camera intrinsic parameters are derived from the OpenGL projection matrix and can be used to back-project the images to a colored point cloud (Figure 3). This is natively supported in ROS, e.g. via the RViz `DepthCloud` plugin or via the `rgbd_launch` package. Optionally, the interface can compute and publish the point cloud data directly.

On a discrete GPU (NVIDIA GeForce GTX 1650 Mobile) this will achieve about 27 Hz, while on an integrated GPU (Intel UHD Graphics 630) this will reduce to about 18 Hz.

### 2.4 Human interaction

Developing contact-aware algorithms is a key aspect of future work for the robotics community [40, 41]. Clearly there are safety concerns for HRI tasks. Simulation is a necessary step in any development cycle involving robots. This means the only way a human can interact with the virtual environment is through some interface (e.g. haptic device). To remedy this issue, we have developed a plugin for several human interfaces so that the human can be realized in the virtual environment and also receive virtual feedback from that environment.

**Telepresence** Physical HRI has obvious safety concerns [42, 43], this motivates including the human in the simulation environment. We provide several interfaces to incorporate a human into the

simulation environment. A ROS driver for the Xsens suit (see Section 3.3) and an avatar (top-right image in Figure 2) is provided to incorporate a human into the simulation. Additionally, several ROS drivers for haptic devices are provided that enable virtual forces to be rendered to the human.

**Control mapping** Teleoperation requires the human to interact with the system via an interface. The signals from the device need to be mapped to a control space, the choice of mapping however is non-trivial [44, 45, 46]. We provide ROS nodes in the `operator_node` package that take raw interface signals as input and maps them to a control space of the users specification. Two options are provided that are typical in robotics applications: (1) Several systems utilize joysticks, often the scaled value of a joystick axis defines velocity in certain dimensions [47, 48]. We provide `scale_node.py` that appropriately orders and scales the interfaces axes. (2) Cartesian/task space control for teleoperation is common, e.g. [44, 45, 49, 6, 33]. The individual axes of the interface is often in the range $[-1, 1]$. If we scaled the joystick axes, as before, then the magnitude of the maximum velocity is non-uniform for all interface states. We provide `isometric_node.py` that ensures the maximum velocity magnitude is isometric.

**Logging signals** There are several advantages for an intermediary node mapping driver signals to operator commands. First, modularity allows the user to easily swap out interfaces/mappings to compare modes. Second, methods utilizing moving horizon estimation (e.g. [50, 51, 6]) need to track a window of signals. We provide a node that enables this functionality in the `operator_interface_logger` node. Third, such a structure enables collection/comparison of data streams using ROS bags removing the need for extensive post-processing.

## 2.5 Interfacing with real hardware

Interfacing with hardware is straightforward using our framework. Each object can publish/broadcast its state in several formats (e.g. joint states, float arrays, transforms, wrenches). This means the simulator can act, at development time, as the real system. When porting to the real hardware we provide nodes that remap the current system states to the required robot/hardware drivers: see the `remap_joint_state_to_floatarray` and `remap_joint_state` nodes in the `custom_ros_tools` package. Setup for these is as simple as remapping topics in a launch file or enabling a ROS re-mapper. The framework can also visualize the current state of physical robots/objects in PyBullet. This is very useful when debugging software and hardware.

## 2.6 Additional utilities

We provide several utilities to facilitate easy transference of robot controllers (learned or otherwise) from simulation to hardware; these are described below.

**Model Predictive Control** When developing MPC methods [7, 6, 8], it is useful to slowly iterate through individual MPC iterations. Start, stop, and step ROS services are provided that allow the user to easily debug MPC controllers (see `rpbi_controls_node.py` in the `rpbi_utils` package).

**Inverse Kinematics** Inverse kinematics is a key requirement for robotic systems with several established libraries [52, 53, 54, 23, 21]. A developer may want to investigate various IK problem formulations by changing constraints and/or cost terms. Hence, we provide a standardized interface `ik_ros` that allows the user to switch between several IK solvers (local, global, trajectory-based, etc) [53, 52, 31] and also define their own problem formulations. The implementation is extensible so that additional solvers can be easily included. Further, a popular integrated solution for IK, and motion planning is MoveIt [21] which, via ROS, could be accessed by the PyBullet community using our framework. In future work, we plan to link our proposed framework with MoveIt.

**Interpolation** Control often requires smaller time resolution than planning. In this case, interpolation between knot points is necessary. The framework provides the `interpolation_node.py` in the `rpbi_utils` package that performs interpolation of planned trajectories.

**Time-sync ROS with PyBullet** Synchronizing time between processes and a simulated time can be useful. We provide an option so that the user can synchronize the ROS clock with PyBullet's simulation time.

**Safe robot operation** Safely operating robots is of paramount importance - especially when conducting HRI experiments. We provide a package `safe_robot` that ensures safe robot motions acting as a guard between target and commanded states - every target is checked (e.g. link and joint position/velocity limits, and self-collision) prior to being commanded on the real system.

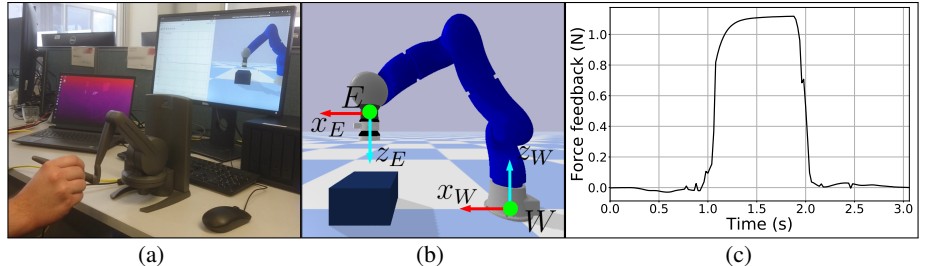

| (a) | (b) | (c) |

Figure 4: A human interacting with a virtual world. (a) The experimental setup where the human interacts with the virtual world using a haptic device. (b) The Kuka LWR robot in PyBullet about to interact with a static collision object including the coordinate frames (the frames are not presented to the user). (c) The force-feeback evolution that is rendered to the human via the haptic device.

## 3 Use-cases

This section describes four use-cases highlighting the features of the proposed framework: (i) human interaction with a virtual world, (ii) learning from demonstration using dynamic movement primitives, (iii) full-body telepresence, and (iv) hardware realization. The code for running examples (i) and (ii) has been made fully open source. We plan to make the driver for the Xsens suit, used in (iii), open source along with an example.

### 3.1 Human interaction with virtual worlds

Developing robust learning and control techniques in contact-rich scenarios utilizing human input requires the human to interact with the virtual world. In order to prototype contact-rich machine learning and optimization-based methods with realistic interaction sequences requires such a haptic interface and a simulator for generating realistic force-feedback.

In this use-case, we present the user with a haptic interface, a simulated Kuka LWR robot arm, and a static collision box object (Figure 4a). Using the interface, the user controls a target position defined in the $z_W$ axis that is constrained in the $x_W, y_W$ axes (Figure 4b). The task for the robot is to minimize the distance between the end-effector position and the target while keeping the $z_E$ and $z_W$ axes aligned. The joint motion is generated using the IK features in PyBullet, we interface with this functionality using the ik_ros package described in Section 2.6. A simulated Force-Torque sensor is attached to the robot at the wrist joint. When the end-effector comes into contact with the static collision object, the force measured by the sensor in the $z_E$ axis is rendered to the user via force-feedback. The evolution of the force feedback for a single interaction with the the box object is shown in Figure 4c. Key to this use-case is to demonstrate that realistic contact force feedback (rendered to the user via a haptic device) can be obtained from PyBullet. To run this example, attach a haptic device (3D Systems Touch X), and execute the command `roslaunch rpbi_examples human_interaction.launch`.

### 3.2 Learning from demonstration

Dynamic movement primitives (DMPs) [55] are a widely used mathematical formulation for modelling motor control of biological systems. Over recent years they have become a key component of learning from demonstration [56]. Many packages, examples, and code exists for learning and executing DMPs – several have been integrated in ROS. To highlight the flexibility and potential for learning from demonstration utlizing our framework we leverage a standard ROS package for learning DMPs [57].

The goal in this section is to demonstrate how to learn a DMP from a teleoperated demonstration using our framework. In this use-case, the user interfaces with the system using the keyboard. A Kuka LWR robot arm is presented in PyBullet, and controlled in position control mode (Figure 5). The interface commands $h$ are mapped to end-effector velocity in two dimensions. We use the EXOTica [23] plugin in the ik_ros package to perform inverse kinematics - the box and end-effector states $x_d, u_d$ are saved as the human demonstration. The goal is for the human to demonstrate how to push a box from a starting location $x_0$ to the goal position $x_g$; this behavior is then learned from the human demonstration $u_d$ using a DMP $\widehat{\theta}$. A random starting location for the end effector is chosen and the DMP is used to plan a motion $\widehat{x}, \widehat{u}$. When the DMP is executed, a starting position is chosen randomly. In contrast to customized data collection schemes, with this use-case, we demonstrate that data can be collected using standard ROS tools and processed to a Pandas data frame for analysis. To run this example, open a terminal and execute `roslaunch rpbi_examples lfd.launch`.

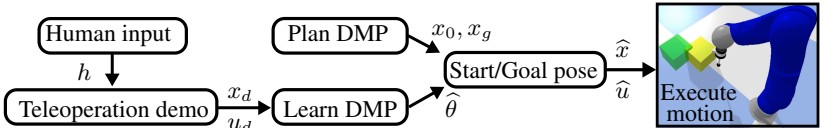

Figure 5: Pipeline for the learning from demonstration use-case.

## 3.3 Full-body telepresence

In this use-case, we show that not only can we setup various teleoperation examples for the development of learning and control, but also we can integrate a realization of a virtual human that can interact with the PyBullet environment. We equip a human with an XSens suit and publish the 3D positions of each human body part into ROS as a transformation. Consecutively, we align each link of the human model in PyBullet with the positions of each body part of the human to realize a full-body virtual-presence of a humanoid figure. The setup is shown via an assistive dressing scenario in Figure 6a, full details of the work can be found in [35]. This use-case demonstrates how a physical HRI task can be validated in simulation, and then easily transferred to the real world.

## 3.4 Integration with hardware and MPC

A key feature of our framework is its ability to easily integrate with robot hardware and develop methods for MPC [6, 8]. In this use-case we highlight this feature of our framework using a pushing task deployed on the Kawada Nextage humanoid robot. The setup is shown in Figure 6c. A typical development cycle for research is to develop first in simulation, and then port the work to the real system. Due to the complexity of robotic systems, data collection, hardware issues, etc., the latter step can be quite time consuming. Our goal during development of the framework was to minimize this difficulty. We developed the facility to remap target joint states to several formats required by real systems in our lab, and several object types that can interface with real sensors (e.g. Vicon, AprilTags) or broadcast transforms (similar to a real object equipped with Vicon markers or AprilTags). The interface makes it simple to swap between testing/prototyping the system in simulation, and deploying on the real system – our demonstration reduces to setting a flag (`real_robot = True` or `False`). Furthermore, this use-case highlights the utility of the time-stepping feature of the interface for developing MPC algorithms, described in Section 2.6. Typically this can only be achieved in simulation, however since our system maps the simulator state to the real robot and has the facility to track objects using online sensing, e.g. we used Vicon, the framework can execute an MPC iteration on the real system by the user clicking a button. This significantly reduces the development time for laborious tasks such as parameter tuning.

## 4 Related work

There are many physics simulators available to the robotics community, e.g. Drake [30], Gazebo [27], Isaac Sim, MuJoCo [29], and PyBullet. However, there does not exist a single simulator that performs best in all desired features for all domains [58]. Table 1 provides a comparison of our framework against other potential open-source solutions. Each simulator is typically developed for a specific purpose or with a certain application in mind, e.g. manipulation, medical, marine, soft robotics, locomotion, etc. Our requirement was a simulator that could reliably model contact and impact for manipulation, as well as haptic interactions. Collins et al. compare the accuracy of manipulation tasks of simulators with respect to real world data [59]. Their comparison showed that PyBullet performed better than V-REP/CopelliaSim and MuJoCo. In terms of contacts, Chung and Pollard show that Bullet (version including the generalized coordinate approach, as in PyBullet) outperformed Dart and ODE for a contact task [60]. A recent study by Acosta [61] compared Py-

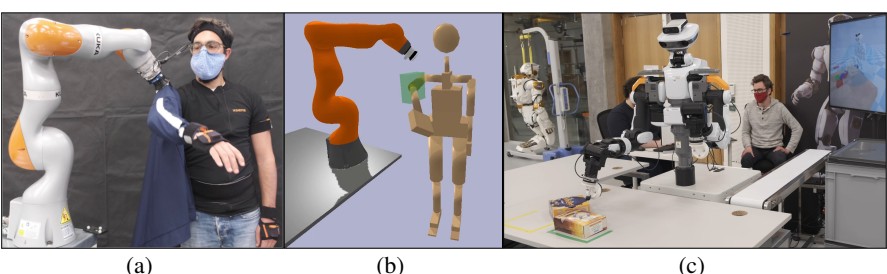

(a)          (b)          (c)

Figure 6: Integrating robot hardware and sensing. (a) and (b) Real-time virtual-presence of a human in the simulation environment during assistive dressing. (a) The configuration of the human is sensed with the Xsens suit. (b) A humanoid figure that corresponds to the human is shown, along with a green *visual object* (Section 2.3.1) used to illustrate the estimated region of the occluded human elbow. (b) The Kawada Nextage humanoid robot performing a pushing task.

| | ROS | Languages | Deform. Obj. | Hardware | HRI | Photo-realistic |
|---|---|---|---|---|---|---|
| Drake | ✗ | C++/Python | ✓ | ✗ | ✗ | ✗ |
| Gazebo | ✓ | C++ | ✗ | ✓ | ✗ | ✗[3] |
| Nvidia Isaac | ✓[4] | Python | ✗ | ✓ | ✗ | ✓ |
| MuJoCo[5] | ✗ | C++/Python | ✓ | ✗ | ✗ | ✗ |
| **ROS-PyBullet** | ✓ | Python | ✓ | ✓ | ✓ | ✗[6] |

Table 1: Comparison of our proposed framework against other potential open source solutions.

Bullet, MuJoCo, and Drake for impact accuracy compared to real world data. PyBullet and Drake were shown to be preferable options for simulating contacts/impacts.

In terms of transference to hardware, Gazebo [27] is arguably one of the most popular simulators in robotics [58, Fig. 2]. Since it's incarnation, Gazebo has supported four backend physics simulators (ODE [27, 28], Dart [39], Bullet, and SimBody). Yet, the current Bullet backend uses maximal coordinate rigid bodies, that is not well suited for robotics (since it allows joint constraint violations). On the other hand, Bullet with the generalized coordinate approach based on Featherstone's algorithm [63] is particularly suitable for robotics (integrated in PyBullet) [60]. Recently, there has been efforts to bridge Nvidia Isaac with ROS/Gazebo. However, as discussed by Gonzalez-Badillo et al. [64] it was found that PhysX outperformed Bullet for assemblies involving simple objects, while Bullet performed better with complex objects. In summary, we argue that PyBullet, with the generalized coordinate is one of the best engines for contact and dynamics modeling and haptic interactions, which is also popular in the learning community (e.g. [65, 66, 67]).

## 5 Limitations

We implemented the framework using Python, a popular programming language with numerous libraries—making it suitable for our goals of easy prototyping and extensiblity. Despite Python being unsuitable for real-time systems, the robotics community has broadly adopted it as the language for implementing robotic experiments. Furthermore, we have found no latency issues in all of our experimental setups, running at frequencies of up to $200\,\mathrm{Hz}$.

Currently, the framework only supports ROS Noetic—thus the only way to run the framework with ROS2 is via the ROS1 bridge [68]. Porting the framework to ROS2 is, at the time of writing this manuscript, in-development.

High quality synthetic image rendering and switching out different physics engines are important features for learning applications. The new Kubric library [62] uses Blender to add photo-realistic rendering to Pybullet. Potential future work of ours is to incorporate the Kubric library and provide a backend interface of our framework to other physics engines, e.g. MuJoCo.

## 6 Conclusions

In this paper, we have proposed a framework for simulating/collecting data for contact-rich manipulation scenarios including: full physics simulation using PyBullet (known for reliable impact/contact modeling), easy transference from simulation to real hardware, integration with the ROS ecosystem,and several teleoperation interfaces and robots. Our focus has been to enable research in contact-rich scenarios involving HRI and haptics, interfacing with virtual worlds. We have chosen a physics simulator that reliably models contact/impact interactions and provide a bridge for the learning comunity to transfer their learned policies to real hardware via ROS. We have specifically designed the implementation to exploit modularity and extensibility and to be highly flexible, easily developed, and be able to interface with common machine learning tools/libraries.

We hope students, researchers, and industry will make use this framework to facilitate the development of their control algorithms and machine learning approaches in scenarios involving contact. The supplementary material contains full documentation. The main code base is open source at github.com/ros-pybullet/ros_pybullet_interface which contains several examples, documentation, and videos. Dependencies are linked to from the documentation and system requirements are detailed. The framework is released under the LGPL license.

---

[3] Possible via additional plugins. [4] Isaac Sim was recently integrated with Gazebo. [5] Until recently MuJoCo required a license. [6] Possible with new Kubric library [62].

## Acknowledgments

This research is supported by The Alan Turing Institute, United Kingdom and has received funding from the European Union's Horizon 2020 research and innovation programme under grant agreement No 101017008, Enhancing Healthcare with Assistive Robotic Mobile Manipulation (HARMONY). This work was supported by core funding from the Wellcome/EPSRC [WT203148/Z/16/Z; NS/A000049/1]. This project has received funding from the European Union's Horizon 2020 research and innovation programme under grant agreement No 101016985 (FAROS project). This research is supported by Kawada Robotics Corporation and the Honda Research Institute Europe. For the purpose of open access, the authors have applied a CC BY public copyright licence to any Author Accepted Manuscript version arising from this submission.

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
