# OpenReview forum: "ROS-PyBullet Interface: A Framework for Reliable Contact Simulation and Human-Robot Interaction"
_robot-learning.org/CoRL/2022/Conference — CoRL 2022 Poster_

### Official Review · Reviewer_VFxS · 2022-07-29

**Originality:** Fair
**Technical Quality:** Very Good
**Clarity Of Presentation:** Good
**Impact:** 3

**Recommendation:**

Strong Accept: I recommend accepting the paper and will argue for my recommendation even if other reviewers hold a different opinion.

**Summary:**

The authors present a bridge between PyBullet and ROS. This empowers researchers to connect an easy-to-work with simulator that’s suitable for simulating contact-rich tasks directly into robotics contexts, where ROS is a dominant middleware. The authors describe the components of their bridge and then present four use cases that highlight the value of the system.

**Issues:**

* Contextualize the work. Provide qualitative comparisons to alternatives. Think about the people you would most like to use your system, and the people who are most likely to come across it, and highlight the value for them.

* Gazebo supported four simulation backends from its earliest days. In fact, in its recent incarnations, DART has become the default. If the other backends are deprecated, or lack a specific feature, point to a notice of the deprecation or documentation of the limitation. You should clarify specifically which version you are referring to, as they have muddied the branding considerably while flip-flopping on their Ignition rename.

**Quality Of The Limitations Section:**

Limitations are addressed clearly

**Reviewer Expertise:**

5: The reviewer is absolutely certain that the evaluation is correct and very familiar with the relevant literature

**Robotics Focus:**

Sufficient demonstration on hardware

**Strengths And Weaknesses:**

Strengths

* Serves a real need in the community. I am aware of several researchers that have independently built subsets of this bridge to suit their needs.

* Well motivated use cases. While simulation solutions oriented toward data generation for learning are the predominant use case in this community, the author’s use cases illustrate some of the main scenarios where simulation is a key part during execution on a real robot system.

Weaknesses

* Limited comparison or contextualization in the landscape of comparable solutions. The paper does not have a related works section, nor does it allude to alternative solutions. The framework tackles a smattering of technical needs (see list on 48-54), which means that readers will come with a variety of comparisons in mind. A table that highlights comparable features and limitations of different options would help.
    * OSRF recently worked with NVIDIA to bridge Gazebo and Isaac Sim (which is PhysX-based). How does this work relate?
    * Readers of CoRL papers may be looking for a contextualization wrt simulators for learning vision-based policies. As the authors call out in 137-140, lots of research is being done that leverages realistic simulated environments. PyBullet doesn’t include a good renderer out of the box. Even discounting the vision use case, nicer renderers are an enabler for work that asks users to interact with the simulation.
    * You may also get readers who are primarily ROS users. They will find the comparisons to Gazebo provided unilluminating, inaccurate, and incomplete. You should highlight why Gazebo can't easily address the needs of certain use cases. In my assessment, it has much more to do with Gazebo requiring the user to know C++ than simply that it's default backend isn't good enough. These readers will also pause at the description of an independent set of kinematics solvers and motion planners, when they are likely most familiar with MoveIt as an integrated solution here. Highlight the benefit you get, be it in ease of use or in capability, compared to simply loading the same URDF into MoveIt and using it's solvers.
    * Necessarily, every effort to enhance robotics simulation platforms will be lacking in some feature, but the onus is on the authors to help prospective users understand how their solution fits in the landscape.

* The motivations for some features may be niche to many readers. There are a couple places (169, 176) where the authors motivate a feature by citing a use case from their own work, indicated by the anonymization of the reference. If there aren’t independent works that have a similar need or use case, it’s an indication that the audience for a particular feature is limited. The authors should consider whether it is worth providing more exposition on the value of these features, or to frame them in an alternate way that more readers will connect with.

**Summary Of Recommendation:**

The authors provide a set of tools which will be valuable either as a reference or even as an immediate aide to researchers. The specific implementation path they took is pragmatic and maintains the strengths of using a common middleware like ROS while enabling some tasks that wouldn’t be possible without a well-behaved simulator in the loop. I think there is room to improve in regards to identifying the key selling points of the system and highlighting contrasts with other options that readers may be considering.

**Update after rebuttal:**

The authors have provided a reasonable qualitative characterization of the landscape of alternatives. I believe this will greatly help readers understand this work and how it fits into the broader landscape. I have raised my recommendation to accept.

---

> ### Author Response · Authors · 2022-08-22
> **Response to Reviewer VFxS of Paper260**
>
> Thank you for your comments. We have addressed them below. Modifications are highlighted in blue text. Please let us know if you have any further feedback and we will incorporate it into our manuscript.
>
> >Limited comparison [..]. The paper does not [..]. A table [..] would help.
>
> >I think there is room to [..] the key selling points [..].
>
> >Contextualize [..]. Provide qualitative [..]. Think [..] and highlight the value for them.
>
> We fully agree with your comments and believe that the additions we made in the manuscript have helped to highlight the value of our framework. We have included a related work section to highlight comparisons with alternatives, and also a table, as you suggested, so potential users can easily assess the usefulness of our framework with respect to their needs.
>
> >OSRF recently [..]Isaac Sim (which is PhysX-based). How does this work relate?
>
> Thank you for pointing us to this, it seems our efforts and this project occurred in parallel. To the extent of our knowledge there is work that we cite in the revised manuscript that supports PyBullet as the favorable simulator when compared to PhysX regarding simulating contacts/impacts. The new related work section discusses this, while the new table also indicates that our framework has features aimed for HRI and Haptics communities, which according to our knowledge Isaac Sim is lacking (apart from XBox controller).
>
> >Readers of CoRL [..] vision-based policies. [..] PyBullet doesn’t [..], nicer renderers are an enabler [..] simulation.
>
> We agree with your point that PyBullet does not include a photo-realistic renderer out-of-the-box. However, this issue can be addressed by extending our framework with the new open-source Kubric library (using photo-realistic renderers via blender, https://pybullet.org/wordpress/index.php/2022/03/21/kubric-a-scalable-dataset-generator/). We have added a pointer to the relevant publication and we emphasize this point in the newly added table.
>
> > You may also get [..] readers who are [..]. [..] comparisons to Gazebo [..]. You should highlight [..].
>
> We agree that one advantage of our work is that it is based on Python, whereas Gazebo requires C++, which we added in the revision. However, the main drawback of Gazebo is that a reliable contact/impact simulator is not currently incorporated as a backend. Gazebo+Bullet uses maximal coordinates (https://www.osrfoundation.org/wordpress2/wp-content/uploads/2015/04/roscon2014_scpeters.pdf), which allows joint constraint violations  (https://pybullet.org/Bullet/phpBB3/viewtopic.php?t=12053), whereas the Featherstone alg. that uses the generalized/minimal coordinates is used in PyBullet. The advantages of the generalized coordinates and of PyBullet are supported by several works that we cite in our updated manuscript.
>
> >Gazebo supported four [..], or [..] point to a notice [..] clarify [..] version, [..].
>
> Indeed, Gazebo provides different backends, however it is shown with the new references that we added that Bullet is one of the best for robotics, manipulation, contact/impact. Regarding the choice of PyBullet or Gazebo+Bullet, as above, PyBullet has clear advantages due to the Featherstone alg.. Whilst there are efforts to extend Gazebo interface to Bullet using the Featherstone alg., this is currently not available and with limited features (https://github.com/gazebosim/gz-physics/pull/373).
> Please see the new related work section that includes the above information.
>
> >These readers [..] set of kinematics solvers [..] familiar with MoveIt [..] Highlight the benefit [..].
>
> Thank you for this comment, we agree with you that MoveIt is a popular solution that most readers would be familiar with. Now, using our framework, the PyBullet community can have access to the features of MoveIt via ROS. Hence, we will consider integrating MoveIt with our framework, and added this as future work.
> Furthermore, we developed ik_ros to interface to several IK solvers that allowed us to modify the IK problem formulation and believe that this could be useful functionality. We have revised sec. 2.6 (IK paragraph) accordingly.
>
> >Necessarily, every [..] will be lacking [..], but the onus is [..]
>
> Thank you for this comment, we hope that the amendments that were made in the contributions, new related work, new table and conclusion sections have improved the manuscript and provide a clear positioning of the framework within the landscape of  similar frameworks.
>
> > The motivations for [..] to frame them in an [..] readers will connect with.
>
> Thank you for pointing these out to us. It is true that the original motivation is based on our own work. However, we believe they fit wider community needs and have included, as you suggested, additional citations that motivate the need of our framework. Please see the revised sec 2.4.
>
> >Limitations are not [..]
>
> We have adjusted the limitations section to be clearer, however, could you please elaborate on what you felt was insufficient here.

---

### Official Review · Reviewer_bKas · 2022-07-30

**Originality:** Good
**Technical Quality:** Fair
**Clarity Of Presentation:** Very Good
**Impact:** 2

**Recommendation:**

Weak Accept: I recommend accepting the paper, but will not argue for my recommendation if the majority of other reviewers have a different opinion.

**Summary:**

This paper presents a ROS-PyBullet Interface bridging between the popular physics simulator PyBullet with Robot Operating System (ROS). Different tools are supported in addition to the native PyBullet and ROS functions such as Inverse Kinematics interface and interfaces for human interaction.

**Issues:**

* This work seems like an integration of multiple pre-existing tools and the contribution section also more or less listed the features of the framework rather than stating the contributions.
* It might be helpful to clearly separate what parts of in this framework are pre-existing and what parts are developed in this work.
* As a paper proposing a system consisting of existing resources (which could be okay), it is very important to demonstrate why it is better than everything else that’s out there. This can be done through experiments comparing with other software, or a dedicated subsection clearly stating the difference and improvement.


**Quality Of The Limitations Section:**

Limitations are addressed clearly

**Reviewer Expertise:**

5: The reviewer is absolutely certain that the evaluation is correct and very familiar with the relevant literature

**Robotics Focus:**

Sufficient demonstration on hardware

**Strengths And Weaknesses:**

**Strengths:**
* The paper clearly outlines the features in ROS-PyBullet and made it very clear what features are supported in the current frame work.
* The framework presented can be a very useful tool for the robotics community. There’s definitely value in developing systems like this.
* Demonstration of hardware integration is helpful.

**Weaknesses:**
* It is not entirely clear what the contribution of this work is after removing the pre-existing modules. In the contribution section, it is claimed that one contribution of the work is the proposed framework that contains different pieces of software, which demonstrates the engineering effort went into the work, but does not speak for the research contribution.
* Related to the previous point, seems like section 2.4, 2.5 and 2.6 are not related to integrating ROS with PyBullet, they are more like standalone ROS nodes (they can be very useful nonetheless).
* It is also not clear how the proposed system helps different projects (Does it achieve things that was not possible before, or does it make things so much easier?) Section 3 listed different use cases for the ROS-PyBullet interface. While it definitely helps to see the example usage, what might be even more helpful is to see how these examples are done differently in the proposed system compared to other systems.


**Summary Of Recommendation:**

ROS and PyBullet are both tools commonly used in the robotics community, and integrating a physics simulation engine with ROS could potentially benefit future research. While the authors clearly achieved the goal of integrating these two tools and even provided a few other tools to make it more user friendly, it is not entirely clear how this integration is fundamentally better than other existing frameworks (or a combination of frameworks). The engineering effort and quality of this work are clear but the research contribution is not well presented.

---

> ### Author Response · Authors · 2022-08-22
> **Response to Reviewer bKas of Paper260**
>
> We thank you for your feedback, and hope the following addresses your concerns. We have made all modifications in the revised manuscript with blue text color.
> Please let us know if you have any further comments/suggestions and we will incorporate them into our revised manuscript.
>
> >It is not entirely clear what the contribution of this work is after removing the pre-existing modules. In the contribution section, it is claimed that one contribution of the work is the proposed framework that contains different pieces of software, which demonstrates the engineering effort went into the work, but does not speak for the research contribution.
>
> >This work seems like an integration of multiple pre-existing tools and the contribution section also more or less listed the features of the framework rather than stating the contributions.
>
> We agree with the comments, hence we rewrote the contribution section to clarify the contributions of our work. In summary these are; (1) a framework for enabling research in contact-rich tasks, (2) a system that enables transference of robot behaviors (learned or otherwise) from the PyBullet simulation environment (popular in the robot learning community) to hardware via ROS. (3) implementation of several HRI interfaces that enable easy interaction of users with virtual environments. (4) several use-cases with different robots to demonstrate the capability/usefulness of the framework.
>
> >Related to the previous point, seems like section 2.4, 2.5 and 2.6 are not related to integrating ROS with PyBullet, they are more like standalone ROS nodes (they can be very useful nonetheless).
>
> Indeed, these sections are not specific to the integration of ROS and PyBullet, they however describe the modules developed that make the framework particularly useful for HRI/Haptics research, and easy transference to the real hardware. Together, the integration of ROS/PyBullet and these modules form our proposed framework.
>
> >It is also not clear how the proposed system helps different projects (Does it achieve things that was not possible before, or does it make things so much easier?) Section 3 listed different use cases for the ROS-PyBullet interface. While it definitely helps to see the example usage, what might be even more helpful is to see how these examples are done differently in the proposed system compared to other systems.
>
> Thank you for your comment, we understand the need for clarifying how we used our framework in each use-case and how it simplified development. Hence, we have included additional information in sections 3.1, 3.2, and 3.3 to highlight how using our framework helps for the development of these use-cases. Regarding section 3.4, we believe that it highlights appropriately the time-stepping feature of our framework that is important for the development/testing/validation of MPC algorithms on virtual and real world systems.
>
> >It might be helpful to clearly separate what parts of in this framework are pre-existing and what parts are developed in this work.
>
> Thank you for your comment, in fig 1, we used color-coding to highlight/separate parts of our framework and we have updated its caption.
>
> >As a paper proposing a system consisting of existing resources (which could be okay), it is very important to demonstrate why it is better than everything else that’s out there. This can be done through experiments comparing with other software, or a dedicated subsection clearly stating the difference and improvement.
>
> Thank you for the comment, we fully agree that this improves the paper. Thus, we have included a related work section, added a table that compares our framework with existing libraries and highlights the key aspects of it in terms of their utility. Also, we updated the conclusion section to further emphasize the key advantages of this system.
>
> >it is not entirely clear how this integration is fundamentally better than other existing frameworks (or a combination of frameworks). The engineering effort and quality of this work are clear but the research contribution is not well presented.
>
> Our need, and we believe it is also a community need, was a way to easily connect a reliable contact/impact simulator with hardware. Several works show PyBullet is able to model contact-rich situations with high accuracy as compared to alternatives. We have added a related work section and table to clarify the advantages/differences of our framework as opposed to alternatives.

---

### Official Review · Reviewer_E5qq · 2022-07-31

**Originality:** Good
**Technical Quality:** Good
**Clarity Of Presentation:** Very Good
**Impact:** 2

**Recommendation:**

Weak Accept: I recommend accepting the paper, but will not argue for my recommendation if the majority of other reviewers have a different opinion.

**Summary:**

This paper presents a framework for generating and simulating multiple scenarios that involve contact data with human demonstrations. The work leverages the pybullet simulation environment and develops robot-centric components for control and sensor measurement. In addition, the work integrates ROS into the the pybullet interface which readily enables the use of human feedback for collecting demonstrations.

**Issues:**

A comparison with Gazebo (and other similar libraries).
Comprehensive overview of other robotics toolbox-libraries, what they provide, and the gap that this fills in.

**Quality Of The Limitations Section:**

Limitations are addressed clearly

**Reviewer Expertise:**

3: The reviewer is fairly confident that the evaluation is correct

**Robotics Focus:**

Sufficient demonstration on hardware

**Strengths And Weaknesses:**

The paper presents several use-cases for their proposed framework. This is presented by the plenty ways to interface and use the presented framework. The breadth of robots and interfaces is very useful when prototyping and developing robotic experiments.

The paper can relate and compare itself more closely with existing simulators that offer similar interfaces with ROS. One example is Gazebo which not only has similar interfaces, but can also choose between contact simulators. Ideally, this work could benefit from allowing other simulators to interface with the ROS environment and the utilities created.

**Summary Of Recommendation:**

The work provides a necessary tool for robotics that can facilitate research progress, especially with human demonstrations. However, there are other tools that claim similar libraries that attain the same contributions. It would strengthen the paper to illustrate where their library stands-out and why this is a preferred way of developing robot simulation utilities.

---

> ### Author Response · Authors · 2022-08-22
> **Response to Reviewer E5qq of Paper260**
>
> We would like to thank you for your comments and feedback, and we have updated our paper according to your recommendations. We have made all modifications in the revised manuscript with blue text color. Please let us know if you have any further comments/suggestions and we will incorporate them into our revised manuscript.
>
> >The paper can relate and compare itself more closely with existing simulators that offer similar interfaces with ROS. One example is Gazebo which not only has similar interfaces, but can also choose between contact simulators.
>
> >However, there are other tools that claim similar libraries that attain the same contributions. It would strengthen the paper to illustrate where their library stands-out and why this is a preferred way of developing robot simulation utilities.
>
> >A comparison with Gazebo (and other similar libraries). Comprehensive overview of other robotics toolbox-libraries, what they provide, and the gap that this fills in.
>
> We found these comments to the point and very useful, hence, we have included a related work section that compares our framework with existing libraries and highlights the key aspects of it in terms of its advantages and its utility. In addition, we have included a table to show diagrammatically how our system relates to other simulators in terms of the features offered.
>
> Specifically, in the case of Gazebo, whilst it provides four different backend physics engines, it is shown in a number of new references that we added in the manuscript that Bullet is one of the best. Please see the newly added related work section.
>
> In addition, one of the backend physics engines is Bullet, however the Gazebo interface only supports maximal coordinates version of Bullet, which is not ideal for robotics as it allows for joint gap (joint constraint violations),  https://pybullet.org/Bullet/phpBB3/viewtopic.php?t=12053). PyBullet, however, uses the Featherstone algorithm (articulated body algorithm), which is using the generalized coordinates (also referred to as minimal coordinates). This is more suitable for robotics, and PyBullet has been shown in several works (which we cite in the revised version of the paper) to also be the favorable simulator for contact/impact tasks. Furthermore, Gazebo requires the user to know C++ whereas PyBullet is in Python (a tool and a language commonly used by the learning robotics community), and PyBullet is very easily installed.
>
> These were the primary reasons why we developed the ROS-PyBullet interface, and we have also included this information in our manuscript to clarify the strengths of our framework. Please see the new related work section, table, and edits in the conclusion section.
>
> >Ideally, this work could benefit from allowing other simulators to interface with the ROS environment and the utilities created.
>
> Indeed, creating a backend for our library to enable alternative simulators, such as MuJoCo, is a very good suggestion and we will consider it as a possible extension of our framework. We have added this point to the limitations section in our revised manuscript and suggested it as future work.

---

### Meta-Review · Area_Chair_JRkk · 2022-08-13

**Recommendation:** Accept (Poster)
**Confidence:** 4

**Metareview:**

Summary:

This paper proposes the ROS-PyBullet interface, a framework that provides a bridge between the reliable contact/impact simulator PyBullet and ROS.
It also provides additional utilities to facilitate human-robot interaction in simulated environments.

Strength:

The paper clearly outlines the features in ROS-PyBullet and made it very clear what features are supported in the current frame work.
Since the simulation solutions oriented toward data generation for learning are the predominant use case in this community, the framework presented can be a very useful tool for the robotics community.

Weakness:

The paper does not have a related works section, nor does it allude to alternative solutions.
It is not entirely clear what the contribution of this work is after removing the pre-existing modules.
Engineering effort is easy to understand; however, it is unclear what the research/academic contributions are.

**Best Paper Nomination:**

No

---

> ### Author Response · Authors · 2022-08-22
> **Response to Meta Review of Paper260**
>
> We thank you for your and all the reviewers’ comments, with which we agree, and the useful feedback. We have incorporated your suggestions into the manuscript revision, the list of changes are as follows.
> * We rewrote the contribution section to clarify the contributions of our work.
> * We added a related work section to better highlight the advantages and differences of our proposed framework as opposed to alternatives.
> * We added a table to visually describe the differences to alternative frameworks.
> * We added a paragraph in the limitations, where we also allude to future work plans.
> * We re-phased the conclusion section to highlight the utility of our framework.
>
> We have made all modifications in the revised manuscript with blue text color.
>
> In summary:
> The main contributions of our work are a system that enables research in contact-rich tasks, easy transference of policies (learned or otherwise) to real hardware via a ROS interface, as well as a number of HRI interfaces that enable human users to interact with virtual worlds. Several works of our own have been enabled by this system, and in the manuscript we demonstrate a community need for the framework.
>
> Furthermore, there currently exists no straightforward way to link PyBullet with hardware and transfer (learned or otherwise) policies onto the hardware. Therefore, we provide a framework to allow easy access (transference) to hardware via ROS. We also provide a number of interfaces to incorporate a human into the simulator via teleoperation and haptic rendering. Furthermore, our framework enables the haptics community to collect data in realistic simulated scenarios for contact-rich tasks.
>
> We decided to use PyBullet, a popular simulator within the robotics and learning community that is reliable for contact/impact situations. While Gazebo is also popular, the backend for Bullet uses maximal coordinates (not such a good option for robotics since it allows for joint gap–joint constraint violation), whereas PyBullet uses the Featherstone algorithm (considered to be the preferable option for robotics). Also, the Bullet backend of Gazebo has received little maintenance within the past several years.
>
> More details on each one of the points mentioned above are provided within the response to the reviewers’ comments.
>
> Please let us know if you have any further comments/suggestions, as we are happy to incorporate them into our revised manuscript.